# Structure and evolution of the Fam20 kinases

Hui Zhang[1,2], Qinyu Zhu[1,2], Jixin Cui[3], Yuxin Wang[1], Mark J. Chen [4], Xing Guo[5], Vincent S. Tagliabracci[6], Jack E. Dixon[3] & Junyu Xiao [1]

The Fam20 proteins are novel kinases that phosphorylate secreted proteins and proteoglycans. Fam20C phosphorylates hundreds of secreted proteins and is activated by the pseudokinase Fam20A. Fam20B phosphorylates a xylose residue to regulate proteoglycan synthesis. Despite these wide-ranging and important functions, the molecular and structural basis for the regulation and substrate specificity of these kinases are unknown. Here we report molecular characterizations of all three Fam20 kinases, and show that Fam20C is activated by the formation of an evolutionarily conserved homodimer or heterodimer with Fam20A. Fam20B has a unique active site for recognizing Galβ1-4Xylβ1, the initiator disaccharide within the tetrasaccharide linker region of proteoglycans. We further show that in animals the monomeric Fam20B preceded the appearance of the dimeric Fam20C, and the dimerization trait of Fam20C emerged concomitantly with a change in substrate specificity. Our results provide comprehensive structural, biochemical, and evolutionary insights into the function of the Fam20 kinases.

[1] The State Key Laboratory of Protein and Plant Gene Research, School of Life Sciences, Peking-Tsinghua Center for Life Sciences, Peking University, 100871 Beijing, China. [2] Academy for Advanced Interdisciplinary Studies, Peking University, 100871 Beijing, China. [3] Department of Pharmacology, University of California, San Diego, La Jolla, CA 92093, USA. [4] Department of Pathology, St. Jude Children's Research Hospital, Memphis, TN 38015, USA. [5] The Life Sciences Institute, Zhejiang University, 310058 Hangzhou, China. [6] Department of Molecular Biology, University of Texas Southwestern Medical Center, Dallas, TX 75390, USA. These authors contributed equally: Hui Zhang, Qinyu Zhu. Correspondence and requests for materials should be addressed to J.X. (email: junyuxiao@pku.edu.cn)

Kinases are molecular activators and signal transducers, and regulate many essential cellular processes by phosphorylating target molecules. The human kinome, for example, encompasses more than 500 kinases that mediate signaling pathways, defects of which are frequently associated with diseases[1]. However, most of the currently characterized kinome members function in the cytoplasm and nucleus to phosphorylate intracellular proteins, despite the fact that phosphorylation of secretory molecules, such as milk casein, had been acknowledged for many years[2].

Recent studies have identified a group of kinases collectively referred to as the "secretory pathway kinases," which are specifically localized to the lumen of the endoplasmic reticulum (ER), Golgi apparatus, and extracellular space[3–5]. Some of these kinases phosphorylate Ser, Thr, or Tyr residues in protein substrates, whereas others possess glycan kinase activities. For example, *Drosophila* four-jointed phosphorylates Ser/Thr residues in the extracellular domains of two atypical cadherins Fat and Dachsous[6]. Vertebrate lonesome kinase (VLK; aka PKDCC/SgK493) phosphorylates a variety of ER resident and extracellular proteins on their Tyr residues[7,8]. In contrast, protein O-mannose kinase (POMK/SgK196), once considered an inactive kinase, phosphorylates a mannose residue to regulate the biosynthesis of alpha-dystroglycan, an essential extracellular matrix glycoprotein required for proper muscle function[9,10]. Some of these kinases are secreted, although it remains unclear whether they can phosphorylate molecules in the extracellular space in a physiological context[8,11,12].

Besides the proteins mentioned above, the "family with sequence similarity 20" (Fam20) proteins also function in the secretory pathway to phosphorylate proteins and proteoglycans. The human genome encodes three Fam20 paralogs: Fam20A, Fam20B, and Fam20C[13]. Remarkably, each has a distinct biochemical activity. Fam20C is the long-sought-after physiological casein kinase that mainly phosphorylates proteins within Ser-x-Glu/pSer motifs[11]. More than 100 proteins are Fam20C substrates, which function broadly in biological processes such as biomineralization, phosphate metabolism, cell adhesion and migration, and cardiac function[11,14–17]. Diminished Fam20C activity causes Raine syndrome, an incurable malady characterized by generalized osteosclerosis and ectopic calcifications[18,19]. Both Fam20A and Fam20B are highly similar to Fam20C in amino acid sequence, but Fam20A lacks an active site residue critical for kinase activity, binds ATP in a catalytically incompetent manner, and is therefore a pseudokinase[20,21]. Fam20A nonetheless promotes the phosphorylation of enamel matrix proteins by forming a complex with Fam20C and stimulating Fam20C activity. Mutations in *fam20a* lead to the dental and renal diseases known as Amelogenesis Imperfecta and Enamel Renal Syndrome[22–24]. Despite these important functions of Fam20C and the Fam20A−Fam20C complex, the structural and molecular basis for regulation of Fam20C by Fam20A are unknown.

In contrast to Fam20A, Fam20B is a bona fide kinase. However, unlike Fam20C, which phosphorylates protein substrates, Fam20B is a glycan kinase like POMK, and its activity is a critical switch during the biosynthesis of chondroitin sulfate (CS) and heparan sulfate (HS) proteoglycans[25,26]. In fact, Fam20B and POMK represent the only two known glycan kinases in animal cells. Proteoglycans are unique macromolecules of the cell surface and major constituents of the extracellular matrix. They are fundamental to a wide spectrum of physiological processes such as adhesion, growth and differentiation, receptor-ligand interactions, and microbial infections[27–29]. The biosynthesis of CS and HS proteoglycans requires the formation of a tetrasaccharide linker, established by the sequential actions of xylosyltransferase

(XylT), galactosyltransferase I (GalT-I), galactosyltransferase II (GalT-II), and glucuronyltransferase I (GlcAT-I) (Supplementary Fig. 1). During this process, Fam20B recognizes the initiator Galβ1-4Xylβ1 disaccharide and phosphorylates the xylose residue at the C2 hydroxyl position[25], and this phosphorylation event is important for priming the activity of GalT-II[26]. Without the priming phosphorylation GalT-II activity is greatly reduced, and formation of the tetrasaccharide linker and subsequent elongation of the CS and HS glycosaminoglycan chains are abolished. Ablation of the *fam20b* gene in mice results in embryonic lethality at E13.5, and is associated with severe developmental defects[30]. Tissue-specific deletions of *fam20b* in joint cartilage and dental epithelium cause chondrosarcoma and biomineralization abnormalities such as supernumerary incisors[31,32]. Zebrafish deficient in Fam20B display malformation of cartilage matrix and bone[33]. These phenotypes highlight the importance of Fam20B function in proteoglycan synthesis and related developmental processes. Despite high sequence similarity, the mechanisms by which Fam20B and Fam20C achieve substrate specificity are unknown.

Here, we use a combination of structural biology, biochemistry, and phylogenetics to elucidate the molecular functions and evolutionary relationships of the three Fam20 kinases. First, we show that Fam20C activation requires the formation of an evolutionarily conserved homodimer, or heterodimer with Fam20A. Compared to Fam20C itself, Fam20A has a more efficient Fam20C-binding surface and is a specialized Fam20C-allosteric activator. We also reveal the mechanism by which Fam20B recognizes its substrate by solving the crystal structure of a Fam20B ortholog in complex with the Galβ1-4Xylβ1 disaccharide. Our phylogenetic analyses suggest that the monomeric Fam20B xylosylkinase activity likely emerged first in evolution, and the evolutionary change in Fam20C substrate specificity correlated with dimer formation. Collectively, these results provide comprehensive insights into the function of this unique and biomedically important family of kinases and shed light on their evolutionary history.

## Results

**Structure of the human Fam20A–Fam20C complex**. To elucidate the molecular basis of how Fam20A regulates Fam20C activity, we determined the crystal structure of the human Fam20A−Fam20C complex (Table 1). The structure reveals that Fam20A and Fam20C form a reversed face-to-face heterodimer (Fig. 1a). The Fam20A−Fam20C interface buries ~1000 Å², or ~5% of the solvent-accessible surfaces from each molecule. At the heart of the interface are interactions between the Kβ3-Kα3 loop (we use "K" to denote the kinase core, and "N" to denote the N-terminal segment) and the N-lobe insertion domain (Kβ5-Kβ7) of each molecule (Supplementary Fig. 2). In addition, residues from the Kβ8-Kα6 loop in the C-lobe of each protein (Leu365$^A$, Lys413$^C$; superscripts A and C indicate human Fam20A and Fam20C, respectively), as well as from the Kβ1-Kβ2 insertion of Fam20A (Ile214$^A$) also contribute to the interaction. Specifically, Ile214$^A$, Phe251$^A$, Phe252$^A$, Ile255$^A$, and Leu365$^A$ form a continuous hydrophobic surface patch on Fam20A that docks onto Fam20C residues including Phe354$^C$, Pro357$^C$, Tyr364$^C$, and Thr373$^C$ (Fig. 1b). Phe306$^A$ and Pro309$^A$ form hydrophobic interactions with Phe300$^C$, and Phe306$^A$ also forms a cation−π interaction with Lys413$^C$. Lys324$^A$ uses its aliphatic side chain to pack against Tyr369$^C$, and its main chain carbonyl group to form a hydrogen bond with His375$^C$. Tyr327$^A$ forms hydrophobic contacts with Phe299$^C$, and a hydrogen bond with Tyr305$^C$. Other hydrogen bonds present at the Fam20A−Fam20C interface include Asp250$^A$-Ser356$^C$, Asp250$^A$-Asn360$^C$, Ser308$^A$-Asp298$^C$,

**Table 1 Data collection and refinement statistics**

| | Fam20A−Fam20C | Fam20A−drFam20C | drFam20C | hmFam20 | hmFam20 with Gal-Xyl |
|---|---|---|---|---|---|
| **Data collection** | | | | | |
| Space group | $P\,2_1\,2_1\,2_1$ | $P\,4_1\,2_1\,2$ | $P\,2_1$ | $P\,2_1\,2_1\,2_1$ | $P\,2_1\,2_1\,2_1$ |
| Cell dimensions | $a = 78.469$ Å, | $a = 110.059$ Å, | $a = 114.925$ Å, | $a = 80.221$ Å, | $a = 80.601$ Å, |
| | $b = 192.747$ Å, | $b = 110.059$ Å, | $b = 135.988$ Å, | $b = 123.222$ Å, | $b = 123.580$ Å, |
| | $c = 226.669$ Å | $c = 461.698$ Å | $c = 219.971$ Å; | $c = 144.393$ Å | $c = 143.936$ Å |
| | | | $\beta = 90.495°$ | | |
| Resolution (Å) | 3.30 | 3.55 | 3.45 | 2.20 | 2.85 |
| $R_{merge}$ | 0.160 (0.815) | 0.351 (1.665) | 0.235 (0.883) | 0.101 (0.995) | 0.216 (1.484) |
| $R_{pim}$ | 0.078 (0.415) | 0.109 (0.546) | 0.143 (0.543) | 0.029 (0.298) | 0.062 (0.445) |
| $I/\sigma I$ | 10.0 (2.25) | 11.5 (2.0) | 8.0 (2.2) | 28.8 (2.9) | 14.8 (2.0) |
| Completeness (%) | 99.7 (100.0) | 100.0 (100.0) | 99.5 (99.0) | 99.7 (95.3) | 99.8 (99.7) |
| Multiplicity | 5.0 (4.7) | 11.4 (10.5) | 3.7 (3.6) | 12.8 (10.9) | 12.8 (11.6) |
| Wilson B-factor | 57.0 | 62.0 | 60.1 | 30.6 | 52.2 |
| **Refinement** | | | | | |
| Reflections used in refinement | 47,610 (2769) | 31,137 (1357) | 88,731 (8732) | 67,332 (4770) | 32,120 (2363) |
| Reflections used for $R_{free}$ | 2159 (125) | 1556 (68) | 2003 (199) | 2000 (142) | 1606 (118) |
| $R_{work}/R_{free}$ | 0.233/0.267 | 0.262/0.300 | 0.225/0.279 | 0.180/0.211 | 0.200/0.217 |
| No. of non-hydrogen atoms | | | | | |
| Protein | 12,888 | 13,508 | 37,919 | 5685 | 5665 |
| Ligands | 124 | 124 | 0 | 56 | 122 |
| Solvent | 0 | 0 | 0 | 545 | 0 |
| Protein residues | 1579 | 1655 | 4594 | 706 | 709 |
| B-factors | | | | | |
| Protein | 68.52 | 61.95 | 54.36 | 41.5 | 49.3 |
| Ligands | 82.38 | 72.14 | | 90.2 | 79.9 |
| Solvent | | | | 44.4 | |
| R.m.s deviations | | | | | |
| Bond lengths (Å) | 0.004 | 0.003 | 0.004 | 0.007 | 0.004 |
| Bond angles (°) | 0.73 | 0.66 | 1.01 | 1.23 | 1.00 |
| Ramachandran | | | | | |
| Favored (%) | 89.61 | 90.23 | 88.10 | 96.8 | 96.7 |
| Allowed (%) | 10.01 | 9.41 | 10.69 | 3.2 | 3.2 |
| Outliers (%) | 0.39 | 0.37 | 1.21 | 0 | 0.1 |

Each dataset was collected from a single crystal. Values in parentheses are for highest-resolution shell

Asn312[A]-Asp298[C], Tyr321[A]-Lys413[C], and Thr325[A]-Glu374[C]. Together, these interactions ensure the specific binding between Fam20A and Fam20C.

To verify the functional relevance of the Fam20A−Fam20C heterodimer, we generated several Fam20A mutants based on the structural observations above, including F251A/F252A, I255E, F306A/P309G, and L365D. These residues were chosen because they are involved in making hydrophobic contacts with Fam20C (Fig. 1b), and we predicted that disrupting them would inhibit Fam20A−Fam20C interaction. We then tested the ability of these Fam20A mutants to bind and activate Fam20C. All of them exhibit greatly reduced interaction with Fam20C as shown by Maltose Binding Protein (MBP) pull-down experiments (Fig. 2a). They also exhibit a substantially diminished capacity to activate Fam20C in vitro when assayed against the Fam20C substrate enamelin (ENAM 173-277) (Fig. 2b). To test the function of these mutants to activate endogenous Fam20C in cells, we analyzed V5-immunoprecipitates from the conditioned medium of $^{32}$P-orthophosphate-labeled U2OS cells expressing V5-tagged osteopontin (OPN) and HA-tagged Fam20A. OPN is a well-characterized Fam20C substrate and its phosphorylation is established as direct readout of endogenous Fam20C activity[11]. Expression of wild-type (WT) Fam20A increases OPN phosphorylation (Fig. 2c), in agreement with our previous results[20]. In contrast, expression of F251A/F252A and F306A/P309G has no effect. Furthermore, WT Fam20A can partially restore the activities of two missense Fam20C mutants identified in Raine patients, Fam20C-G379E and Fam20C-G280R, as judged by the

change in electrophoretic mobility of V5-tagged OPN on SDS-PAGE (Fig. 2d)[20]. The F251A/F252A and F306A/P309G mutants of Fam20A failed to activate the Fam20C Raine mutants, whereas I255E and L365D showed only partial activation. The various defects of these Fam20A mutants confirm that the dimer interface observed in the crystal structure is critical for Fam20A to bind Fam20C and regulate its activity.

**Fam20C proteins function as evolutionarily conserved dimers.** We also generated several mutants of Fam20C that we predicted would disrupt dimerization with Fam20A, including F299A/F300A, F354A/P357G, H375Y, and E374S/H375T (Fig. 1b). Interestingly, these mutants show substantially decreased basal kinase activity when assayed in the absence of Fam20A (Fig. 3a). All these mutant proteins were secreted from insect cells as efficiently as WT Fam20C, suggesting that the mutations do not severely disrupt protein folding. These observations raised the intriguing possibility that Fam20C may also be regulated by homodimerization. Indeed, purified Fam20C oligomerizes in a concentration-dependent manner in solution, and is mainly a dimer at ~1 mg per ml, as shown by analytical ultracentrifugation (Supplementary Fig. 3A) and size exclusion chromatography experiments (Supplementary Fig. 3B). In contrast, Fam20C-F354A/P357G, the mutant that displays the lowest kinase activity (Fig. 3a), is largely monomeric (Supplementary Fig. 3B).

The human Fam20C protein crystallized but the crystals diffracted poorly. To gain structural insights into the Fam20C

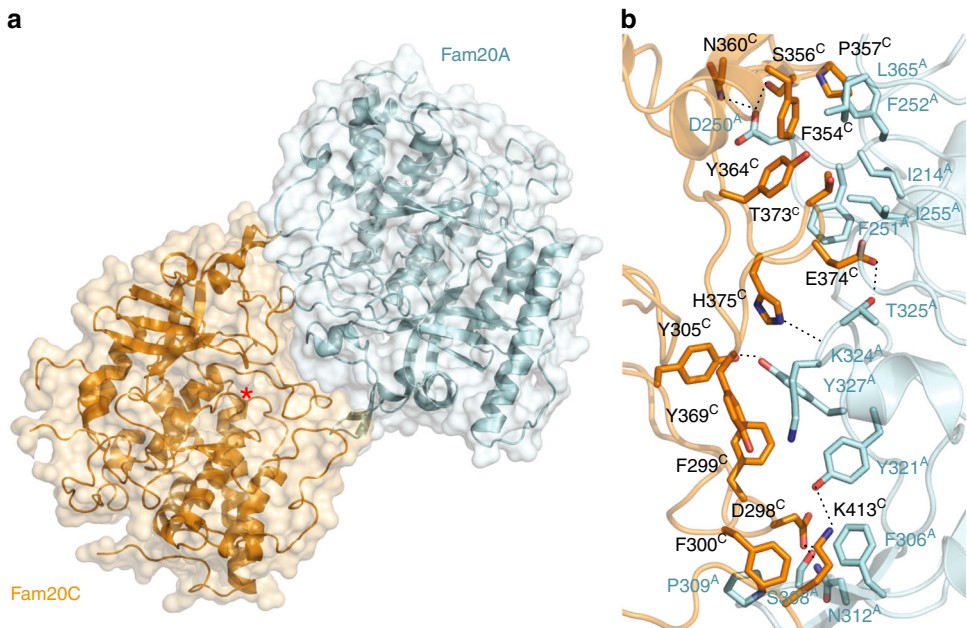

**Fig. 1** Crystal structure of the human Fam20A−Fam20C complex. **a** Overall view of the human Fam20A−Fam20C heterodimer. The active site of Fam20C is indicated with an asterisk. **b** Detailed view of the Fam20A−Fam20C interface. Superscripts A and C denote residues in Fam20A and Fam20C, respectively. Hydrogen bonds are indicated by dashed lines

homodimer, we crystallized a Fam20C ortholog from zebrafish (*Danio rerio*, drFam20C), which is 86% identical (94% similar) to human Fam20C in the kinase domain, and determined its structure (Table 1). There are 12 molecules of drFam20C in the crystal asymmetric unit, which organizes into six pairs of homodimers similar in overall structure to the Fam20A−Fam20C heterodimer (Fig. 3b, Supplementary Fig. 4A). All the residues that participate in the drFam20C dimer interface are conserved in human Fam20C (Fig. 3c, Supplementary Fig. 2), including residues corresponding to Phe299[C], Phe300[C], Phe354[C], Pro357[C], Glu374[C], and His375[C] (Phe275, Phe276, Phe330, Pro333, Glu350, and His351 in drFam20C), mutations of which reduce Fam20C activity (Fig. 3a). These analyses, together with the biochemical results described above, suggest that the human Fam20C protein probably forms a similar homodimer, and that dimer formation is critical for allosterically activating kinase activity.

Interestingly, a homologous homodimer is also present in the crystal lattices of ceFam20, formed by two symmetry-related molecules (Fig. 3d, Supplementary Fig. 4A). ceFam20 is a Fam20C ortholog in *Caenorhabditis elegans*[34], which does not have Fam20A. Consistently, ceFam20 is a dimer in solution, and ceFam20-F260A/P263G (analogous to human Fam20C-F354A/P357G, Supplementary Fig. 2) is monomeric (Supplementary Fig. 3C). Importantly, the kinase activity of this mutant is also greatly reduced (Fig. 3e). Together, these results demonstrate that the Fam20C proteins function as evolutionarily conserved dimers.

A comparison of the drFam20C homodimer with the Fam20A−Fam20C heterodimer suggests that Fam20C uses an almost identical set of residues to interact with itself and Fam20A (Figs. 1b, 3c, Supplementary Fig. 4B). However, Fam20A contains several unique features that make it a more efficient Fam20C-interactor. For example, Ile214[A], Ile255[A], and Leu365[A] are unique to Fam20A (Supplementary Fig. 2), and contribute to the formation of an optimized hydrophobic surface for interacting with Fam20C (Fig. 1b, Supplementary Fig. 4B). Lys324[A] and Tyr327[A] are also unique to Fam20A and mediate additional interactions with Fam20C in the middle of the Fam20A−Fam20C interface (Supplementary Fig. 2, Fig. 1b, Supplementary Fig. 4B). Fam20C is expressed ubiquitously, whereas Fam20A expression is

restricted to specific tissues, particularly ameloblasts and the lactating mammary gland[20,30,35] where high Fam20C activity is likely required for phosphorylating enamel matrix and milk proteins. Thus, we propose that the Fam20C homodimer plays a "house-keeping" function, whereas Fam20A is employed when Fam20C activity is in high demand.

**The Fam20A−Fam20C tetramer**. Our previous biochemical data suggested that Fam20A and Fam20C form a heterotetramer consisting of two molecules each of Fam20A and Fam20C[20,21]. However, this state is not present in the human Fam20A−Fam20C crystal. In an attempt to address this discrepancy, we solved the structure of the Fam20A−drFam20C complex that is crystallized in a different space group (Table 1). A four-leaf clover-shaped tetramer is clearly present in this structure (Fig. 4a), formed by two Fam20A−drFam20C heterodimers similar to the human Fam20A−Fam20C heterodimer shown in Fig. 1a. A helix in the N-terminal region of drFam20C (Nα2[drFam20C]), which is conserved in human Fam20C (Supplementary Fig. 2), plays a pivotal role in mediating the tetramer formation.

In light of this Fam20A−drFam20C tetramer structure, we realized that the absence of a human Fam20A−Fam20C tetramer in the crystal is also due to an unexpected obstructing ATP molecule, besides crystal packing effects. Both Fam20A−Fam20C and Fam20A−drFam20C only crystallized in the presence of ATP, likely because ATP significantly stabilizes Fam20A[21]. In both structures, an ATP molecule (ATP-1) is bound to Fam20A in an inverted orientation, as we previously reported[21]. In the human Fam20A−Fam20C crystal, an additional ATP (ATP-2) is bound to Fam20A perpendicularly to ATP-1 (Supplementary Fig. 5A), and is incompatible with tetramer formation because it would sterically occlude the Nα2 helix in Fam20C (Supplementary Fig. 5B). In agreement with this observation, excess ATP converts the human Fam20A−Fam20C tetramer to a dimer in solution (Supplementary Fig. 5C).

To assess the functional importance of the tetramer, we designed two Fam20A mutants: E299G/I300S and K129A/R132A/R136A. The E299G/I300S mutations create a site for

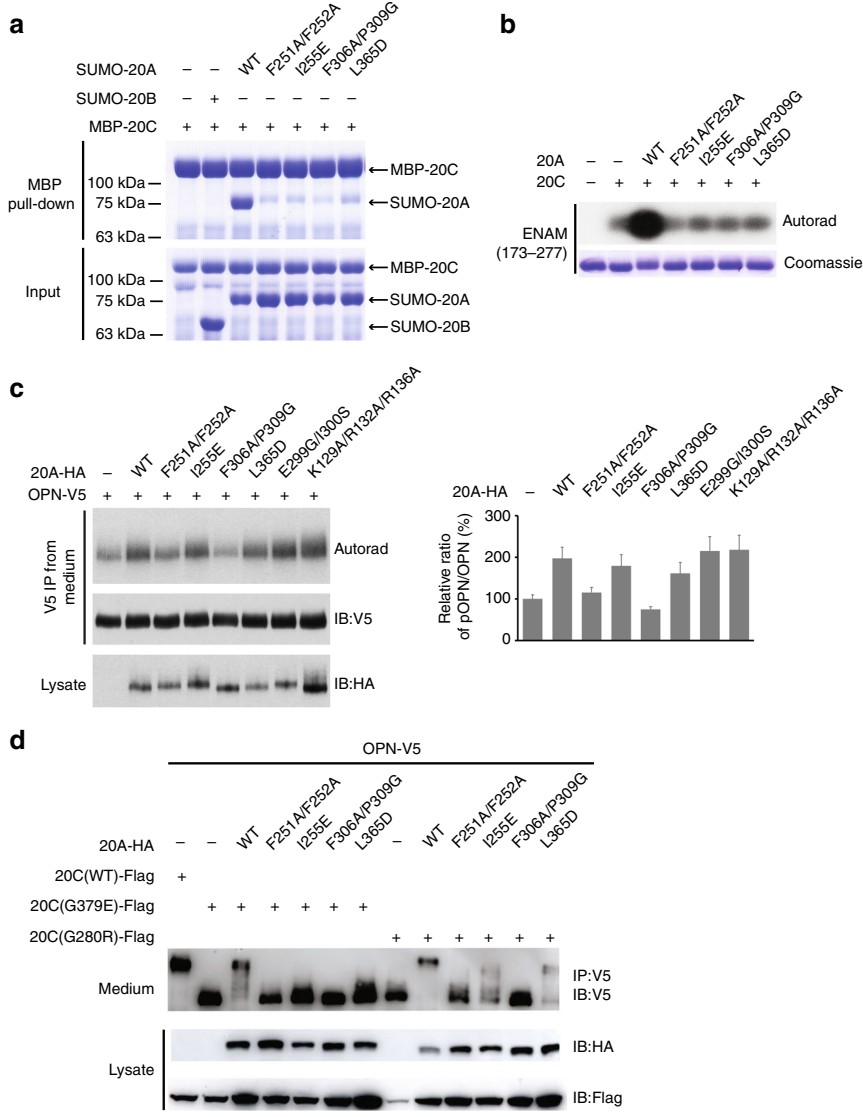

**Fig. 2** The dimer interface is critical for Fam20A to bind and activate Fam20C. **a** Fam20A mutants display reduced interaction with Fam20C. His₆-SUMOstar-Fam20A (WT and mutants), His₆-SUMOstar-Fam20B, and His₆-MBP-Fam20C were individually isolated from the condition media of insect cells using Ni-NTA affinity purification. They were mixed as indicated (input), and MBP pull-down was performed using amylose resin and analyzed by SDS-PAGE and Coomassie staining. **b** Fam20A mutants exhibit substantially diminished abilities to activate Fam20C. Phosphorylation of human ENAM (173–277), as indicated by the incorporation of $^{32}$P from [γ-$^{32}$P]ATP, was examined by SDS-PAGE and autoradiography. **c** Fam20A-F251A/F252A and Fam20A-F306A/P309G are unable to enhance OPN phosphorylation catalyzed by endogenous Fam20C. OPN-V5 was co-expressed with WT or mutant Fam20A-HA as indicated. Cells were metabolically labeled with $^{32}$P orthophosphate before OPN-V5 was immunoprecipitated from the conditioned media. Total protein and $^{32}$P incorporation were detected by immunoblotting and autoradiography. The relative phosphorylation level of OPN was represented by the ratio of $^{32}$P autoradiography and V5 immunoblot intensity, normalized to the ratio from cells without Fam20A transfection (two replicates, error bars representing standard deviation). **d** Fam20A mutants are not able to promote the activities of two Fam20C mutants associated with Raine syndrome, G280R and G379E. C-terminally V5-tagged OPN (OPN-V5), C-terminally Flag-tagged Fam20C (Fam20C-Flag, WT or mutants), and C-terminally HA-tagged Fam20A (Fam20A-HA, WT or mutants) were co-expressed in U2OS cells as indicated. Secreted OPN-V5 was immunoprecipitated from conditioned media and detected by immunoblotting. OPN phosphorylation was evaluated by its mobility shift on SDS-PAGE

N-linked glycosylation (Asn298-Gly299-Ser300). Given that Asn298[A] is at the center of the Fam20A−drFam20C tetramer interface (Fig. 4a), the bulky glycan addition is predicted to hinder tetramer formation. Lys129[A], Arg132[A], and Arg136[A] probably interact with the Nα2 helix in the tetramer (Fig. 4a), and are also involved in binding to ATP-2 in the Fam20A−Fam20C structure (Supplementary Fig. 5A). These two mutants reduced and abolished human Fam20A−Fam20C tetramer formation, respectively, as shown by size exclusion chromatography (Supplementary Fig. 5D). Nevertheless, both mutants activated Fam20C at levels comparable to WT Fam20A in vitro

(Fig. 4b) and in cells (Fig. 2c). These data suggest that while a Fam20A−Fam20C heterodimer is required for Fam20A to enhance Fam20C activity, further assembly of a tetramer is dispensable for activation. The functional significance of ATP-2 remains unclear, and we cannot rule out the possibility that it is a crystallization artifact.

**Substrate recognition mechanism of Fam20B.** Human Fam20C and Fam20B are 42% identical within the kinase domain yet Fam20B is a glycan kinase that regulates proteoglycans

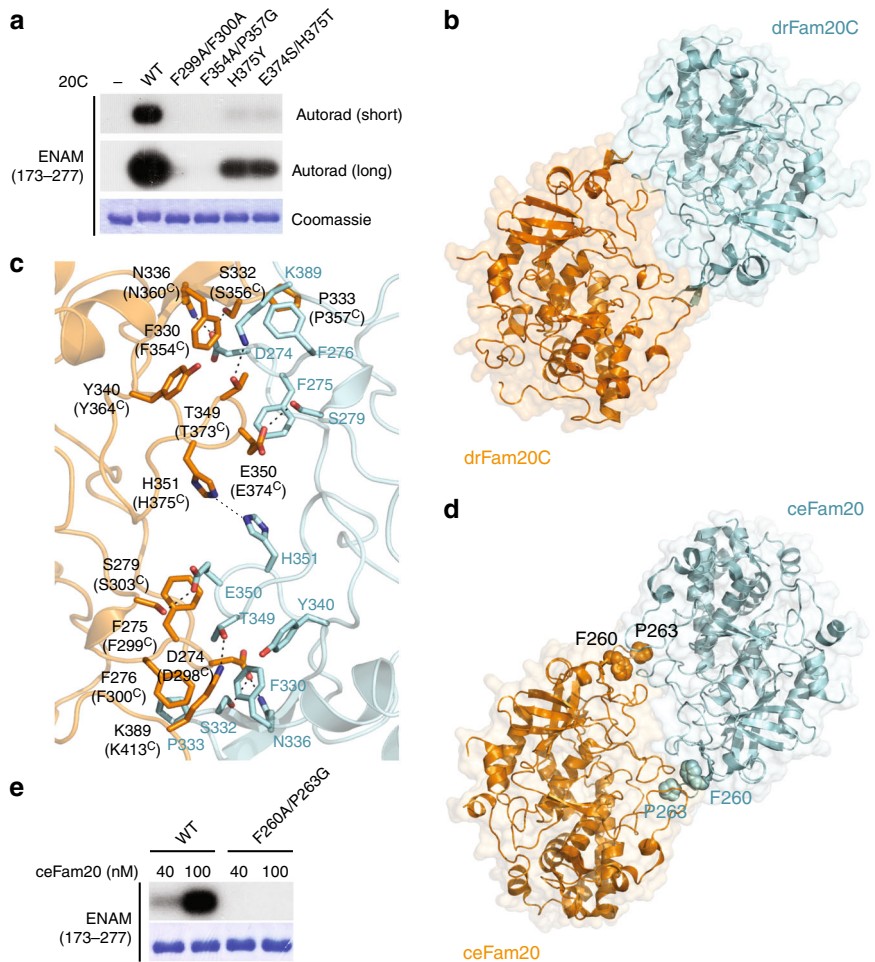

**Fig. 3** Evolutionarily conserved dimer formation is required for Fam20C activity. **a** Fam20C mutants display reduced kinase activity. **b** The crystal structure of drFam20C reveals a homodimer structure resembling the human Fam20A−Fam20C heterodimer. **c** Detailed view of the drFam20C homodimer interface. All the residues involved in drFam20C homodimer formation are conserved in human Fam20C (the corresponding residues in human Fam20C are indicated in brackets). **d** Reexamination of the crystal structures of ceFam20 reveals a homodimer structure present in the crystal lattices of two different crystal forms (PDB entries 4KQA and 4KQB). Phe260 and Pro263 are highlighted. **e** The kinase activity of ceFam20-F260A/P263G is diminished

biosynthesis. To gain insights into the evolution of the Fam20 kinases, we searched annotated animal genomes by Position-Specific Iterative Basic Local Alignment Search Tool (PSI-BLAST)[36] for Fam20 homologs. Several lower animals encode a single Fam20 protein (Fig. 5a). We previously demonstrated that the single Fam20 homolog in *C. elegans* is a Fam20C-like kinase[34] despite its previous annotation as Fam20B[13]. In contrast, the single Fam20 proteins in basal animals such as sponge (*Amphimedon queenslandica*, aqFam20) and hydra (*Hydra magnipapillata*, hmFam20) are apparently Fam20B-like because they exhibit robust activity for the Galβ1-4Xylβ1 disaccharide (Gal-Xyl hereafter, Fig. 5b), and do not phosphorylate protein substrates such as OPN, casein, and enamelin (Fig. 5c). In fact, xylose phosphorylation within the tetrasaccharide linker region of CS proteoglycans was detected in hydra[37] prior to the discovery of the Fam20B xylosylkinase activity[25]. Similarly to human Fam20B, aqFam20 and hmFam20 are monomers (Supplementary Fig. 3D).

To further understand the structural difference between Fam20B and Fam20C, we sought to determine the crystal structure of Fam20B. Although human Fam20B can be purified to homogeneity, it failed to crystallize despite extensive trials. hmFam20 crystallized readily, but the initial crystals diffracted poorly. Treatment of the purified hmFam20 with endoglycosidase

F3 to remove the majority of the N-linked glycans significantly improved crystal quality and diffraction. The hmFam20 structure was subsequently determined at 2.2 Å resolution (Table 1). The overall structure of hmFam20 resembles that of Fam20C, and can be superimposed onto ceFam20 with a root-mean-square deviation of 1.5 Å over 342 aligned Cα atoms. There are four disulfide bonds in hmFam20 (Cys207-Cys222, Cys212-Cys215, Cys267-Cys340, Cys341-Cys400), which all align with the four disulfide bonds in Fam20C.

To explore the substrate recognition mechanism of Fam20B, we also determined the structure of hmFam20 in complex with Gal-Xyl (Table 1, Fig. 6a). Although AMP-PNP, a nonhydrolyzable ATP analog is also present in the crystal solution, electron densities can only be clearly discerned for the adenosine moiety. The phosphate groups display weak electron densities and are not modeled. A comparison of the hmFam20 structure with the Mn/ADP-bound ceFam20[34] reveals the ATP-binding site is highly conserved between Fam20B and Fam20C (Fig. 6a). Asp152 occupies the position of Glu213$^{ceFam20}$ and forms an ion pair with Lys133. Although Glu218$^{ceFam20}$, instead of Glu213$^{ceFam20}$, aligns with the ion pair Glu in canonical kinases (such as Glu91 in PKA), Glu213$^{ceFam20}$ is nevertheless important for the kinase activity of Fam20C[34], and Fam20A is a pseudokinase partly

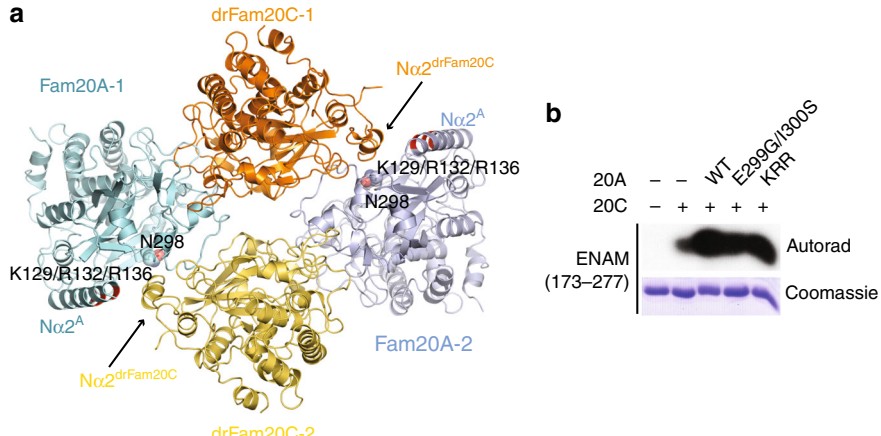

**Fig. 4** The Fam20A−Fam20C tetramer. **a** The crystal structure of Fam20A−drFam20C reveals a four-leaf clover-shaped tetramer formed by two Fam20A−drFam20C heterodimers. The side chain of Asn298$^A$ is shown. The positions of Lys129$^A$, Arg132$^A$, and Arg136$^A$ in the Nα2$^A$ helix are highlighted in red. The side chains of these three residues are not clearly visualized in the structure. **b** The E299G/I300S and K129A/R132A/R136A mutants of Fam20A activate Fam20C at levels comparable to WT Fam20A in vitro

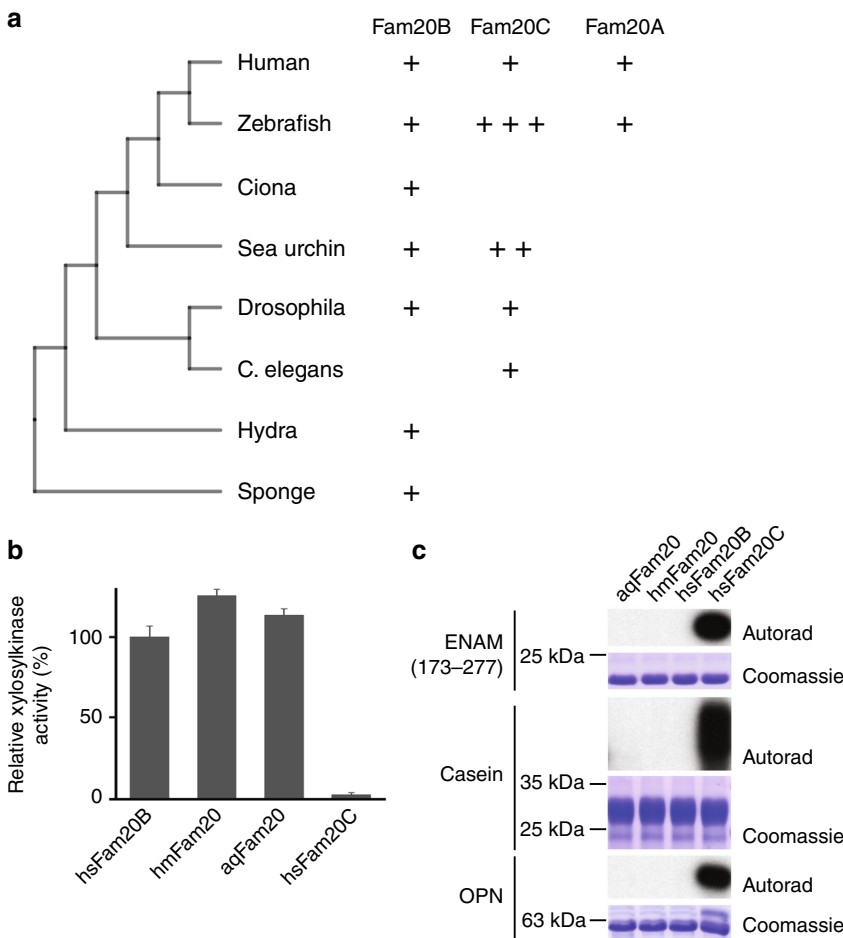

**Fig. 5** Phylogenetic analyses of the Fam20 proteins. **a** A diagram depicting the presence of different Fam20 proteins in representative species. **b** The Fam20 proteins from sponge (aqFam20) and hydra (hmFam20) phosphorylate the Gal-Xyl disaccharide with efficiencies similar to human Fam20B (hsFam20B). Error bars represent the standard deviation of three independent experiments. **c** Only human Fam20C phosphorylates enamelin, casein, and OPN, as revealed by $^{32}$P autoradiography

because it contains a Gln instead of an acidic Asp/Glu at this position[20,21]. The Gal-Xyl is accommodated next to the ATP-binding site, in a pocket formed by Thr114, Gln115, Tyr148, Glu149, Gly150, Tyr151, Tyr214, Tyr253, His301, and Lys321 (Fig. 6b). Thr114 forms a hydrogen bond with the C3 hydroxyl group of the Xyl. Gln115 forms hydrogen bonds with both the C3 hydroxyl group of the Xyl and the C5 hemiacetal oxygen in the Gal. Gly150 uses the main chain amine and carbonyl groups to

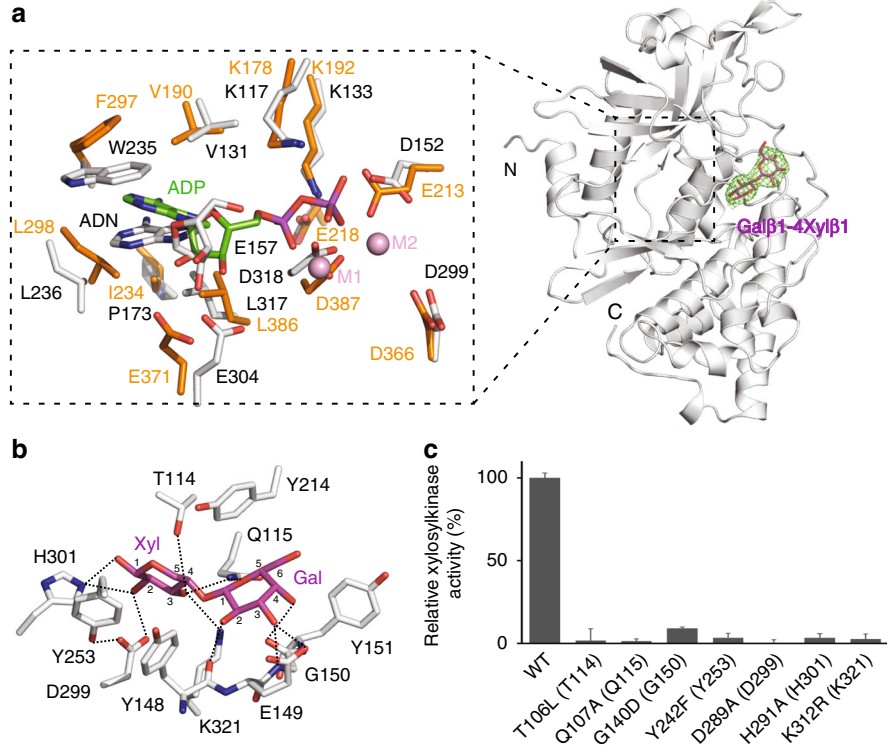

**Fig. 6** Substrate recognition mechanism of Fam20B. **a** The structure of hmFam20 in complex with Gal-Xyl. hmFam20 is shown as white ribbons. The $F_o-F_c$ difference electron density map (3.0 $\sigma$) calculated before Gal-Xyl was modeled is shown as a green mesh, revealing the presence of the disaccharide. A close view of the ATP-binding site is displayed on the left, with the Mn/ADP-bound ceFam20 structure (shown in orange) superimposed for comparison. ADN, adenosine. The two Mn ions in ceFam20 are labeled as M1 and M2. **b** An enlarged image of the disaccharide-binding pocket showing the detailed molecular interactions between hmFam20 and Gal-Xyl. **c** Human Fam20B mutants display reduced or abolished kinase activity. The amino acids in brackets indicate the corresponding residues in hmFam20. Error bars represent the standard deviation of three independent experiments

form two hydrogen bonds with the C3 and C4 hydroxyl groups of the Gal. Tyr253 forms a platform to support the Xyl together with Tyr148, and also forms a hydrogen bond with Asp299 to help position this critical residue. Asp299 aligns with the H/YRD Asp in protein kinases, and forms a hydrogen bond with the C2 hydroxyl oxygen of the Xyl. This coordination supports the function of Asp299 as the catalytic base, similar to the H/YRD Asp[38,39]. His301 forms hydrogen bonds with the C1 and C2 hydroxyl groups of the Xyl. Lys321 hydrogen bonds with the C3 hydroxyl group of the Xyl. Importantly, these residues are highly conserved in Fam20B (Supplementary Fig. 2), and mutations of the equivalent residues in the human Fam20B protein significantly diminish kinase activity (Fig. 6c). These data corroborate our structure results, and demonstrate that Fam20B use an evolutionarily conserved mechanism to specifically recognize the Gal-Xyl disaccharide as a substrate.

**Fam20C dimer emerged with a change in substrate preference.** The hmFam20 structure enabled us to consider how Fam20C may have emerged in higher animals. Inspection of the hmFam20 structure (Fam20B-like) and the ceFam20 structure (Fam20C-like) reveals three regions in Fam20C that lead to a change in substrate preference. First, the Kβ3-Kα3 loop of Fam20C contains a D/N-F/H-F-Y-F-S/T-D motif (Clash1; Figs. 7a, b). Within this motif, the F-Y-F (Phe207[ceFam20]-Tyr208[ceFam20]-Phe209[ceFam20], Fig. 7c) cluster together by strong hydrophobic/π interactions, and the invariant Asp (Asp211[ceFam20]) is anchored to an Arg (Arg214[ceFam20]). In the context of these structural restraints, the F-S/T would clash with Gal-Xyl (Phe209[ceFam20]-Ser210[ceFam20],

Fig. 7c). Second, the Kβ6-Kβ7 loop of Fam20C contains a C-D/S-Y-Y-C motif (Clash2). The two Cys that form a disulfide bond (Cys273[ceFam20]-Cys277[ceFam20]) are spaced by three residues, as opposed to two residues found in Fam20B (Fig. 7a). Consequently, a Tyr is pushed down and would hinder the binding of Gal-Xyl (Tyr275[ceFam20], Fig. 7c). Finally, an Arg in the Kβ10-Kα9 loop of Fam20C (Clash3) replaces a critical Lys in Fam20B (Fig. 7a), and the more extended side chain of the Arg would prevent disaccharide binding (Fig. 7c). Indeed, changing the corresponding Lys to Arg in human Fam20B (K312R, Fig. 6c) completely abolishes kinase activity.

Interestingly, some of these changes correlate with the capacity of Fam20C to dimerize. The D/N-F/H-F and the S/T residues in the D/N-F/H-F-Y-F-S/T-D motif are involved in the homodimer interface of ceFam20 and drFam20C (Asn205[ceFam20]-Phe207[ceFam20], Ser210[ceFam20]; Asp274[drFam20C]-Phe276[drFam20C], Ser279[drFam20C]; Supplementary Fig. 2; Fig. 3c). The corresponding residues in human Fam20C are also involved in forming the Fam20A−Fam20C heterodimer (Asp250[A]-Phe252[A], Ile255[A]; Asp298[C]-Phe300[C]; Fig. 1b). Similarly, one of the Tyr in the C-D/S-Y-Y-C motif is located in the ceFam20 dimer interface (Tyr276[ceFam20], Supplementary Fig. 2), and is exploited by human Fam20A and Fam20C for the formation of the Fam20A−Fam20C heterodimer (Tyr321[A], Tyr369[C]; Fig. 1b). Therefore, it appears that emergence of the capacity for Fam20C to form dimers was concomitant with its divergence from the Fam20B glycan kinase during evolution. Changes in other places, especially the appearance of the F-x-S-P motif in the N-lobe insertion, further stabilize the Fam20C homodimers (Phe260[ceFam20]-Pro263[ceFam20], Phe330[drFam20C]-Pro333[drFam20C], Supplementary Fig. 2, Fig. 3c) and the Fam20A−Fam20C heterodimer (Phe306[A]-Pro309[A], Phe354[C]-Pro357[C], Fig. 1b).

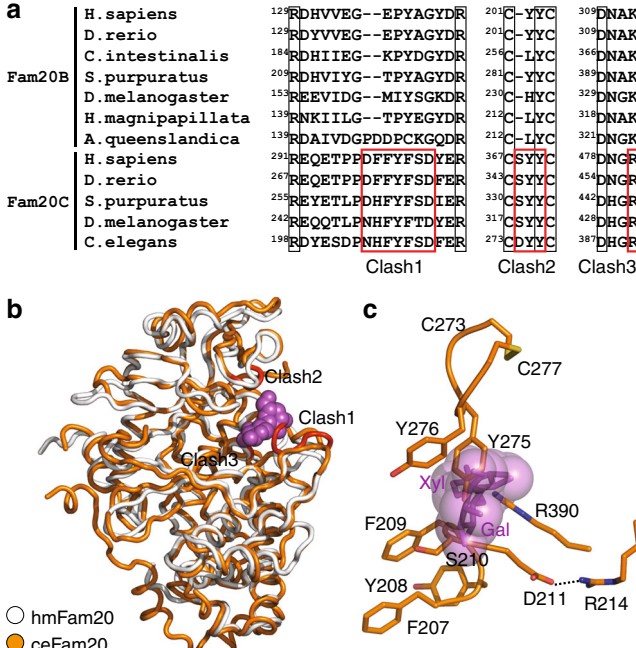

**Fig. 7** Changes in Fam20C lead to the loss of glycan kinase activity. **a** Sequence alignment of Fam20B and Fam20C family members in three regions that are critical for substrate selectivity. Residues that are incompatible with disaccharide binding in Fam20C are highlighted using red boxes and labeled Clash1, Clash2, and Clash3. **b** ceFam20 (orange) is superimposed onto hmFam20 (white) in the Gal-Xyl bound structure. Gal-Xyl is shown as magenta spheres. The three clash regions in ceFam20 are highlighted in red. **c** A close view of the clash regions. Gal-Xyl is shown as magenta sticks enclosed in its van der Waals surface. F209, S210, Y275, and R390 in ceFam20 would clash with Gal-Xyl

## Discussion

The Fam20 proteins are recently identified kinases that regulate essential cellular processes by phosphorylating proteins and proteoglycans. In this study, we provide mechanistic insights into how these kinases function. We first show that Fam20C achieves a catalytically productive state when it is engaged in a reversed face-to-face dimer configuration with another molecule of Fam20C or its pseudokinase paralog Fam20A. This allosteric mechanism for Fam20C activation, governed by dimerization, is remarkably similar to several canonical kinase systems, including the PKR, RAF, and EGFR kinases, in which an "activator" kinase/ pseudokinase modulates the conformation of the "receiver" kinase and leads to its activation[40–46], despite the fact that the dimers of these kinases are formed in very different manners (Supplementary Fig. 6). The parallels between Fam20A, KSR, and Her3 are especially uncanny. KSR and Her3 are paralogs of the RAF and EGFR kinases, respectively. Both lack certain residues that are normally required for catalysis and are thus considered pseudokinases[1]. While they display some kinase activities using unconventional reaction mechanisms[47–49], their functions as allosteric regulators of RAF and EGFR are well-established[50,51]. From a broader perspective, pseudoenzymes are prevalent not only among kinases but other enzyme families as well, and it is becoming increasingly clear that these proteins are actively involved in a range of cellular functions, often by mediating protein–protein interactions[52,53]. Many of them pair with an active counterpart to modulate its function. Among the 13 secretory pathway kinases or kinase-like proteins in humans identified to date, more than half remain poorly characterized, and most of them have not been shown to possess kinase

activity[16]. Sequence analyses indicate that critical catalytic residues are missing in some of them, such as DIA1R/CXORF36[54]. It remains to be determined whether they have unique active site structures like POMK and phosphorylate as-yet unknown substrate molecules, or rather act as pseudoenzymes and have functions independent of catalytic capacity.

All Fam20C structures obtained so far are involved in a homodimer configuration or heterodimer with Fam20A, and likely reflect an active conformation. The absence of a Fam20C structure in the monomeric/inactive state leaves our understanding of its activity-switching mechanism incomplete. Interestingly, we noticed that two regions in Fam20A undergo large conformational changes when engaged with Fam20C: the Kβ3-Kα3 loop and the Kβ8-Kα6 loop (Supplementary Fig. 7). In the Fam20A alone structure[21], these two loops interact with another Fam20A molecule to form a Fam20A homodimer (Supplementary Fig. 7A) that is distinct from the Fam20A−Fam20C heterodimer and Fam20C homodimer. When Fam20A is in complex with Fam20C, these two loops move significantly to mediate the heterodimer interaction (Supplementary Fig. 7B). Interestingly, in this state, the two loops in Fam20A become very Fam20C-like (Supplementary Fig. 7C). Whether Fam20C can undergo similar conformational changes is unknown, but the sequence similarity between Fam20A and Fam20C (Supplementary Fig. 2) suggest that the corresponding regions in Fam20C could be inherently dynamic. The Kβ3-Kα3 loop of Fam20C contains Glu306$^C$ (equivalent to Glu213$^{ceFam20}$ mentioned above) that is essential for catalysis, while several positively charged amino acids that are potentially involved in substrate binding reside in the Kβ8-Kα6 loop[20,34]. Conformational changes of these regions would thus directly impact Fam20C activity. Besides Fam20A, small molecules such as sphingosine and several sphingolipids are known to also activate Fam20C[55,56], but the underlying molecular mechanism is currently unknown. It remains to be understood whether these small molecules promote the homo-dimerization of Fam20C, or act to stabilize the conformation of the two loops discussed above. Fingolimod, a structural analog of sphingosine and a clinically approved drug for treating multiple sclerosis, also stimulates Fam20C kinase activity[55]. Further understanding the mechanisms by which Fam20C is regulated may facilitate the design of better therapeutics for patients carrying non-lethal Fam20C mutations.

Our work also shed light on the evolution of the Fam20 kinases. We show that the Fam20B xylosylkinase activity and specificity are evolutionarily conserved, and can be traced back further than the Fam20C protein kinase in the animal kingdom (Fig. 5). Sponges are considered the oldest animal phylum[57], and we find that the single Fam20 protein in sponge is Fam20B-like. Interestingly, although the sponge glycans are structurally distinct from typical CS and HS glycosaminoglycans[58,59], homologs of XylT, GalT-I, GalT-II, and GlcAT appear to exist in *A. queenslandica* (NCBI protein IDs: XP_011404143, XP_003383478, XP_003384004, XP_003386699). This suggests that *A. queenslandica* is capable of producing the tetrasaccharide linker with a phosphorylated xylose residue. Hydra is the most basal animal that contains classic CS and HS glycosaminoglycans[37]. The single Fam20 protein in hydra is also Fam20B-like, which likely plays a similar role in regulating proteoglycan synthesis as its ortholog in higher animals. In contrast, the single Fam20 in *C. elegans* is a protein kinase like Fam20C. Consistently, only unphosphorylated xylose was detected in the tetrasaccharide linker of *C. elegans* CS proteoglycan[60]. A close examination based on the information gained in this study suggests that the Fam20 proteins in other nematodes are also Fam20C-like, although some are still annotated as the Fam20B glycosaminoglycan xylosylkinase (such as the Fam20 in the parasitic worm *Strongyloides ratti*,

NCBI ID: CEF70981.1). Most higher animals encode both Fam20B and Fam20C in their genomes. For example, fruit fly (*Drosophila melanogaster*) has one copy each of Fam20B and Fam20C[14]. Although their functions are not yet characterized, xylose phosphorylation within the tetrasaccharide linker region of CS and HS proteoglycans has also been demonstrated in *Drosophila*, and this activity can probably be ascribed to Fam20B[60]. Sea urchin (*Strongylocentrotus purpuratus*) contains one Fam20B (NCBI protein ID: XP_011661062) and two Fam20C (XP_791445, XP_011661640). However, Fam20C appears to have been lost in tunicates such as *Ciona intestinalis* (Fig. 5a). Although the incomplete nature of the *Ciona* genome makes this observation less certain, another tunicate, *Oikopleura dioica*, also seems to have only Fam20B, and no Fam20C. In contrast, both Fam20B and Fam20C are present in other invertebrate chordates such as *Branchiostoma* (XP_002593700, XP_002606109) and *Saccoglossus* (XP_006822578, XP_002731199). Among vertebrates, the expansion of Fam20C is found in fish. For example, zebrafish has three copies of Fam20C (XP_009298166, XP_688892, XP_001345757) (Fig. 5a). Fam20A is also first found in fish[13]. In conjunction with our biochemical and structural observations, these phylogenetic analyses suggest that the monomeric Fam20B glycan kinase might have preceded the appearance of Fam20C in evolution. The appearance of Fam20C protein kinase is associated with dimer formation, in which the two protomers stabilize each other's conformations. Fam20A is probably derived from Fam20C given their close sequence similarity, and has lost bona fide kinase activity but acquired a specific Fam20C-activator function.

In summary, we have solved a series of crystal structures, including that of the human Fam20A−Fam20C complex and a Fam20B ortholog in complex with its disaccharide substrate. Our results have provided a complete set of structural templates for the future study of this family of kinases, offered a deeper understanding of their regulation and substrate specificity, and revealed a unique example of protein evolution.

## Methods

**Cell culture**. Sf21 and High Five cells, originally purchased from Invitrogen, were maintained using a non-humidified shaker at 27 °C in the SIM SF medium and the SIM HF medium (Sino Biological Inc.), respectively. U2OS cells, originally purchased from ATCC, were grown in a humidified incubator with 5% $CO_2$ at 37 °C in Dulbecco's modified Eagle's medium (DMEM) containing 10% fetal bovine serum (FBS) with 100 μg per ml penicillin/streptomycin (GIBCO). All cell lines were tested to be free of mycoplasma by the standard PCR method. Identity of the cell lines were frequently assessed by their morphological features.

**Protein expression and purification**. Primers used in this study were listed in Supplementary Table 1. The *Amphimedon queenslandica Fam20* (aqFam20) was cloned from a cDNA library (a kind gift from Prof. Bernard Degnan, University of Queensland). The codon-optimized *hmFam20* gene was synthesized (GenScript, Supplementary Table 2). The *Danio rerio Fam20C* gene (drFam20C) was cloned from a zebrafish cDNA library (a kind gift from Prof. Bo Zhang, Peking University). The human *Fam20A*, *Fam20B*, and *Fam20C* genes were applied as previously described[11].

For expression in insect cells, DNA fragments encoding aqFam20 (residues 31–418), hmFam20 (residues 55–415), hsFam20B (residues 55–402), hsFam20A (residues 69–529), hsFam20C (residues 63–584), and drFam20C (residues 124–560) were cloned into the psMBP2 vector and expressed as $His_6$-MBP fusion proteins[61]. Bacmids were generated using the Bac-to-Bac system (Invitrogen). Baculoviruses were generated and amplified using the sf21 insect cells. For protein production, High Five cells were infected at a density of 1.5–2.0 million cells per ml for 48 h. The conditioned media were collected by centrifugation at $200 \times g$, concentrated using a Hydrosart Ultrafilter (Sartorius), and exchanged into the binding buffer (20 mM Tris-HCl, pH 8.0, 200 mM NaCl). The recombinant proteins were then isolated using Ni-NTA affinity purification and eluted with 20 mM Tris-HCl, pH 8.0, 200 mM NaCl, 250 mM imidazole. Mutations were introduced into plasmids by a PCR-based method, and the mutant proteins were purified as above for the WT protein.

To prepare protein samples for crystallization, purified $His_6$-MBP-hsFam20C, $His_6$-MBP-hsFam20A, or $His_6$-MBP-drFam20C were digested with TEV protease

to remove the N-terminal $His_6$-MBP fusion tag. Untagged hsFam20C, hsFam20A, drFam20C were then purified by anion exchange chromatography (Resource Q), eluted using a 50–500 mM NaCl salt gradient in 20 mM Tris-HCl, pH 8.0; followed by size exclusion chromatography (Superdex increase 200), eluted in 10 mM HEPES, pH 7.5, 100 mM NaCl. $His_6$-MBP-hmFam20 was digested with TEV protease and endoglycosidase F3 simultaneously to remove $His_6$-MBP and N-linked glycans. Untagged and deglycosylated hmFam20 was then purified by cation exchange chromatography (Resource S), eluted using a 50–500 mM NaCl salt gradient in 20 mM MES, pH 6.5; followed by size exclusion chromatography (Superdex 200 16/600), eluted in 10 mM HEPES, pH 7.0, 150 mM NaCl.

**Crystallization**. All crystals were grown at 20 °C using the hanging drop or sitting drop vapor diffusion method. To crystallize the Fam20A−Fam20C complexes, purified Fam20A was incubated with human Fam20C or drFam20C in the presence of 0.25 mM ATP on ice for 1 h, and then passed through a size exclusion column (Superdex increase 200). The purified complexes in 20 mM HEPES pH 7.5, 100 mM NaCl, and 0.25 mM ATP were then concentrated to about 6 mg per ml. The human Fam20A−Fam20C complex was crystallized in 2% 1,4-Dioxane, 0.1 M Bicine, pH 9.0, and 9% Polyethylene glycol 20,000. The Fam20A−drFam20C complex was crystallized in 11% 2-Propanol, 0.1 M Sodium citrate tribasic, pH 5.0, and 8% Polyethylene glycol 10,000. drFam20C alone crystals were obtained in 0.2 M Ammonium citrate tribasic, 0.1 M Imidazole, pH 7.0, and 18% Polyethylene glycol monomethyl ether 2000. Apo hmFam20 was crystallized in 1.5 M $(NH_4)_2SO_4$, 0.1 M BIS-TRIS propane, pH 7.0, and 3% dextran sulfate. The disaccharide-bound hmFam20 was crystallized in 1.8 M $NaH_2PO_4$/$K_2HPO_4$, pH 6.9, 10 mM $MgCl_2$, 5 mM AMP-PNP, and 2 mM Galβ1-4Xylβ1-O-benzyl. All crystals were transferred into the crystallization solution plus 20% glycerol and flash-frozen in liquid nitrogen for data collection.

**Data collection and structure determination**. The crystal diffraction data were collected at the Shanghai Synchrotron Radiation Facility (beamline BL17U) and the National Facility for Protein Science Shanghai (beamline BL19U). The data were processed using HKL2000 (HKL Research). All structures were solved by the molecular replacement method using Phaser[62], with the ceFam20 and hsFam20A structures (PDB IDs: 4KQA and 5WRR[21,34]) as search models. The structural model was then manually built using Coot[63] and refined using Phenix[64]. Five percent randomly selected reflections were used for cross-validation[65]. Final structures were validated with the MolProbity program in Phenix and the wwPDB server[66].

**In vitro kinase assay**. The kinase assays were performed as previously described[20,26]. Fam20C kinase assays were performed in 50 mM HEPES, pH 7.0, 60 mM NaCl, 10 mM $MnCl_2$, 0.5 mg per ml BSA, and 100 μM [γ-$^{32}$P]ATP (specific activity 5000 cpm per pmol) using OPN, casein, or human ENAM (173–277) as substrate. The kinase reactions were initiated by the addition of recombinant hsFam20C (40 nM), ceFam20 (40 nM), or a combination of hsFam20C (40 nM) and hsFam20A (40 nM); incubated for 10 min at 30 °C; terminated by the addition of SDS-PAGE buffer plus 20 mM EDTA; and then boiled. The reaction mixtures were then separated by SDS-PAGE and visualized by Coomassie staining. $^{32}$P incorporation was detected by autoradiography. Uncropped images of gels and blots are shown in Supplementary Fig. 8.

Fam20B xylosylkinase assays were carried out in 50 mM HEPES, pH 7.5, 10 mM $MnCl_2$, 100 μM Galβ1-4Xylβ1-O-benzyl, 100 μM [γ-$^{32}$P]ATP (specific activity 500 cpm per pmol), and 4 μg per ml various Fam20 proteins at 30 °C for 30 min. Reactions were terminated with 40 mM EDTA and 10 mM ATP, and loaded onto Sep-Pack C18 cartridges (Waters) pre-equilibrated with 0.2 M $(NH_4)_2SO_4$. The columns were washed with 2 ml of 0.2 M $(NH_4)_2SO_4$ three times, and the disaccharides were eluted with 1 ml methanol. Incorporated radioactivity was measured by liquid scintillation counting (Tri-Carb 2810TR, PerkinElmer).

**Analytical ultracentrifugation**. Sedimentation velocity experiments were carried out on a Beckman XL-I Analytical Ultracentrifuge. Purified hsFam20A-hsFam20C (~0.8 mg per ml, 400 μl) in 20 mM HEPES, pH 7.5, 100 mM NaCl, with or without 1 mM ATP was spun at $50,310 \times g$, 20 °C for 10 h, and the 280 nm absorbance data were recorded. Data analysis was performed using SEDFIT[67].

**Cell-based Fam20C activation assay**. For co-expression experiments, U2OS cells were grown in a six-well plate format to ~40–50% confluency. Cells were co-transfected with 0.75 μg pCCF-Fam20C (WT or mutants), 0.75 μg pCDNA-Fam20A-HA (WT or mutants), and 1.5 μg pcDNA-OPN-V5 using 6 μl X-tremeGENE-9 (Roche). Conditioned media were harvested 40–48 h later, centrifuged at $500 \times g$ for 5 min to remove the cell debris, further cleared at $8000 \times g$ for 10 min, and then used for immunoprecipitation. V5-tagged proteins were immunoprecipitated using anti-V5 rabbit (Millipore, AB3792) and protein-G agarose (Pierce, 20399), washed three times with PBS, and eluted with SDS loading buffer. To analyze the total intracellular proteins (total extract), cells were washed with PBS, lysed with 200 μl of SDS loading buffer, boiled, and then used for immunoblotting. Proteins were separated by SDS-PAGE, transferred to PVDF membranes, blocked in 5% milk, and probed with anti-FLAG M2 (mouse, Sigma, F3165), anti-V5 (mouse, Invitrogen, R960-25), and anti-HA (mouse, sigma,

H9658) antibodies. Detection was performed by enhanced chemiluminescence using an Amersham Imager 600.

For metabolic radiolabeling experiments, U2OS cells grown in the six-well plates were co-transfected with 4 μg of pCDNA-OPN-V5 and 10 ng pCDNA-Fam20A-HA (WT or mutants). Two days after transfection, metabolic labeling was started by replacing the medium with phosphate-free DMEM containing 10% dialyzed FBS and 1 mCi per ml $^{32}$P orthophosphate (PerkinElmer). After labeling for 8 h, the conditioned medium was collected and the cell debris was removed by centrifugation. V5-tagged proteins were immunoprecipitated from the supernatant and analyzed for protein and $^{32}$P incorporation by immunoblotting and autoradiography.

**Quantification and statistical analysis**. The quantitative values obtained in the figures were analyzed in Excel spreadsheets with the embedded basic statistical functions (mean, standard deviation).

**Data availability**. Atomic coordinates and structural factors have been deposited in the Protein Data Bank with accession codes 5YH3, 5YH2, 5YH0, 5XOM, and 5XOO for Fam20A-Fam20C, Fam20A-drFam20C, drFam20C, hmFam20, and disaccharide-bound hmFam20, respectively. Other data are available from the corresponding author upon reasonable request.

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

## Acknowledgements

We are grateful to the staff of the Shanghai Synchrotron Radiation Facility (beamline BL17U) and the National Facility for Protein Science Shanghai (beamline BL19U) for assistance with X-ray data collection. We thank Kathrein Roper for providing the *A. queenslandica* cDNA library, Rob Steele for validating the *H. magnipapillata* Fam20 sequence, and Xinquan Wang for providing the expression vector of endoglycosidase F3. We thank Carolyn Worby and Sandra Wiley for their critical reading of the manuscript. Special thanks to Chen Song for insightful discussions. The work was supported by the National Key Research and Development Program of China (2017YFA0505200 and 2016YFC0906000) and the National Natural Science Foundation of China (31570735) to J.X, and the NIH grant R00DK099254 and Welch Foundation Grant I-1911 to V.S.T. Some of this work was carried out in Jack Dixon's laboratory, and he would like to acknowledge support from the NIH (grants DK 18849 and 18024) and the Howard Hughes Medical Institute.

## Author contributions

Conceptualization, J.X.; Investigation, H.Z., Q.Z., J.C., Y.W., M.J.C., and J.X.; Writing—Original draft, H.Z., Q.Z., and J.X.; Writing—Review and editing, X.G., V.S.T., J.E.D., and J.X.; Funding acquisition, V.S.T., J.E.D. and J.X.; Supervision, J.X.

## Additional information

**Competing interests:** The authors declare no competing interests.

