## [Peer Review File(PDF 215 kb) · Nature Communications]

Reviewers' comments:

Reviewer #1 (Remarks to the Author):

The paper from Zhang and colleagues is from a group of collaborators who have made seminal discoveries in the field of atypical kinase regulation, and most notably recently, that the FAM20C kinase, and its allosteric activator, the FAM20A pseudokinase, control the vast majority of phosphorylation on secreted proteins. Indeed, FAM20C has now been revealed as the physiological casein kinase, so understanding its regulation is of utmost importance, especially given the links between FAM20C and diseases relevant to mineralisation.

This new study contains novel findings pertaining to the mechanism of hyperactivation of FAM20C by FAM20A, through heterodimer/tetramerisation, and also evaluates FAM20C homodimerisation, and how the FAM20B xylose kinase, which regulates an early phosphorylation step of proteoglycan biosynthesis (another extracellular biomolecule), binds to a disaccharide substrate. The findings are wide-ranging, important and will lead to further advances in this rapidly developing field. I am particularly interested that the authors have chosen to bring together proteoglycan and protein phosphorylation by analysing the two related mechanistic processes; this is important in the field, and certainly overcomes the lack of cellular analysis presented in this study. Phylogenetic and structural analysis then come together impressively, leading the authors to hypothesise that the FAM20B monomer xylose phosphorylating activity evolved prior to a dimeric FAM20C protein kinase activity, which thus generated the potential for a heterodimeric FAM20A regulated allosteric complex. The work is appropriately cited, and the data are high quality, both in terms of biochemistry (using kinase assays for FAM20C with the substrate enamel and osteopontin) and in terms of biophysical analysis, with appropriate controls throughout. In particular, a wide-ranging set of mutational studies are employed, with conserved residues identified in zebrafish Fam20C at dimerisation interfaces found to be conserved in human. The ability of FAM20A to activate FAM20C through a 'reversed face-to-face dimer', driven by this conserved hydrophobic patch is likely to be physiologically relevant, especially since FAM20A has the ability to activate mutated (low activity) FAM20C proteins, an impressive feat. The authors conclude that FAM20A and FAM20C collaborate to control FAM20C output, probably based on demand (which will vary at different times, eg. during developmental processes)

Major point:

There is no doubt from previous work that FAM20B/POMK/Sgk196 are glycan kinases. With the interesting evolutionary trajectory proposed, and because the authors have tested osteopontin as a potential FAM20B substrate in a simple assay (Figure 5C), can they also confirm that FAM20B does not phosphorylate enamel (173-277), and more interestingly, what happens if FAM20B is titrated into a mixture of FAM20A/FAM20C or FAM20C or FAM20A alone, in terms of their ability to phosphorylate these protein (or indeed the xylose glycan) substrates?

Minor points:

1) Figures 5B and 6C. No loading for the disaccharide substrate (Galbeta1-4Xylbeta1-O-benzyl) is shown in this assay, presumably because this is challenging and due to the way the assay is performed. However, it would strengthen the paper to explain how the authors actually quantify 1) the concentration of this substrate in the assay, and 2) how the authors confirmed that the substrate was present at the same concentration in each assay.

2) Additionally, how do the authors know which position on the xylose ring is phosphorylated? Is this based on the previously published papers, if so, please clarify.

Transparent review process: This review was submitted by Prof Patrick Eyers, University of Liverpool,

and he was the sole reviewer.

Reviewer #2 (Remarks to the Author):

The authors present structural and biochemical data on the Fam20 protein kinases, an interesting and important family of kinases responsible for the phosphorylation of secreted proteins and proteoglycans. The detailed structural analysis builds on the authors' earlier work and rationalises many of their previous findings; while providing new insight into the function as well as the evolutionary history of the protein family.

The manuscript is carefully prepared, logical and well written, but could benefit from a few minor changes, as detailed bellow:

1. Page 3, line 49: Change "sequence similarity 20" to "sequence similarity of 20 %".
2. Page 4, line 61: Change FAM20A to fam20A.
3. Page 5, line 81: Change FAM20B to fam20B.
4. Page 6, line 105: K β 3-K α 3 nomenclature is not in agreement with traditional kinase secondary structure nomenclature. If this is common nomenclature for the FAM20 family? Please specify in the text or adjust the nomenclature accordingly.
5. Figure 2: Quantification of autoradiography data should be displayed, as a bar graph. This would be useful to more easily compare activity data between the mutants. Furthermore this data should be normalised to account for differing protein levels (Fig. 2C, 2D).
6. Page 9, line 75: Comparison between the Fam20A homodimer and Fam20C heterodimer interface would be clearer if the two interfaces were overlaid on a single, figure (an additional figure added to the SI).
7. Page 10, line 207: E299 and I300 are discussed in the text but are not highlighted in Fig. 4.
8. Page 11, line 233: Is aqFam20 actually a larger protein containing more amino acids compared with the other family members or does it just behave as a larger protein on SDS-PAGE, SEC etc? This should be addressed in the text or at least in Fig. S2 (where aqFAM is not included).
9. There is much discussion on how the active site in Fam20B makes up the dimer interface of Fam20C, rendering it inaccessible to substrate, but not much discussion on where the active site of Fam20C is with respect to Fam20B. A figure of Fam20C and Fam20B overlaid with their respective catalytic sites highlighted, and a few sentences addressing this point would be of interest to the reader.
10. Page 14, line 302: How do the face-to-face dimers of FAM20 kinases compare to other the other pseudo-kinase/kinase heterodimers that are mentioned in the text? An additional supplementary figure should be included to highlight common and diverse features of these dimeric arrangement.
11. Fig. S2: The black box surrounding certain residues is undefined in the figure legend.

Reviewer #1 (Remarks to the Author):

The paper from Zhang and colleagues is from a group of collaborators who have made seminal discoveries in the field of atypical kinase regulation, and most notably recently, that the FAM20C kinase, and its allosteric activator, the FAM20A pseudokinase, control the vast majority of phosphorylation on secreted proteins. Indeed, FAM20C has now been revealed as the physiological casein kinase, so understanding its regulation is of utmost importance, especially given the links between FAM20C and diseases relevant to mineralisation.

This new study contains novel findings pertaining to the mechanism of hyperactivation of FAM20C by FAM20A, through heterodimer/tetramerisation, and also evaluates FAM20C homodimerisation, and how the FAM20B xylose kinase, which regulates an early phosphorylation step of proteoglycan biosynthesis (another extracellular biomolecule), binds to a disaccharide substrate. The findings are wide-ranging, important and will lead to further advances in this rapidly developing field. I am particularly interested that the authors have chosen to bring together proteoglycan and protein phosphorylation by analysing the two related mechanistic processes; this is important in the field, and certainly overcomes the lack of cellular analysis presented in this study. Phylogenetic and structural analysis then come together impressively, leading the authors to hypothesise that the FAM20B monomer xylose phosphorylating activity evolved prior to a dimeric FAM20C protein kinase activity, which thus generated the potential for a heterodimeric FAM20A regulated allosteric complex. The work is appropriately cited, and the data are high quality, both in terms of biochemistry (using kinase assays for FAM20C with the substrate enamel and osteopontin) and in terms of biophysical analysis, with appropriate controls throughout. In particular, a wide-ranging set of mutational studies are employed, with conserved residues identified in zebrafish Fam20C at dimerisation interfaces found to be conserved in human. The ability of FAM20A to activate FAM20C through a 'reversed face-to-face dimer', driven by this conserved hydrophobic patch is likely to be physiologically relevant, especially since FAM20A has the ability to activate mutated (low activity) FAM20C proteins, an impressive feat. The authors conclude that FAM20A and FAM20C collaborate to control FAM20C output, probably based on demand (which will vary at different times, eg. during developmental processes)

Major point:

There is no doubt from previous work that FAM20B/POMK/Sgk196 are glycan kinases. With the interesting evolutionary trajectory proposed, and because the authors have tested osteopontin as a potential FAM20B substrate in a simple assay (Figure 5C), can they also confirm that FAM20B does not phosphorylate enamel (173-277), and more interestingly, what happens if FAM20B is titrated into a mixture of FAM20A/FAM20C or FAM20C or FAM20A alone, in terms of their ability to phosphorylate these protein (or indeed the xylose glycan) substrates?

Author's response:

- (1) We have tested whether Fam20B can phosphorylate other protein substrates including casein and enamel. None of the three Fam20B orthologues (aqFam20, hmFam20, and human Fam20B) can phosphorylate these molecules. The data is presented in the new Figure 5C.
- (2) In a previous study, we have carried out very similar experiments as Dr. Evers suggested. We showed that Fam20B does not promote the protein kinase activity of Fam20C, nor does Fam20A or Fam20C has any effect on Fam20B-catalyzed xylose phosphorylation (Cui, J. et al. *Elife* 4, e06120 (2015); Figure 4B, 4C). We also showed previously that Fam20B does not interact with Fam20C (Cui, J. et al. *Elife* 4, e06120 (2015); Figure 5B, 5C) or Fam20A (Cui, J. et al. *Elife* 6, e23990 (2017); Figure 4B). These results can now be rationalized in light of the structure information: Fam20B does not have the interface required for dimerization with Fam20C/Fam20A.

Minor points:

1) Figures 5B and 6C. No loading for the disaccharide substrate (Galbeta1-4Xylbeta1-O-benzyl) is shown in this assay, presumably because this is challenging and due to the way the assay is performed. However, it would strengthen the paper to explain how the authors actually quantify 1) the concentration of this substrate in the assay, and 2) how the authors confirmed that the substrate was present at the same concentration in each assay.

Author's response:

As Dr. Eyers rightfully pointed out, it is challenging to show the loading for the disaccharide substrate, since it is a small molecule rather than a proteinaceous substrate, therefore we cannot justify the loading by SDS-PAGE. The way we performed these experiments is that we first made a 10x stock solution (1 mM) of the disaccharide, by weighing the right amount and dissolving in water. Then we took out equal volumes from the stock solution and mixed with other assay components (ddH₂O, 10x reaction buffer, 10x ATP stock solution). Finally, we initiated the reactions with the Fam20 proteins. We relied on careful pipetting to make sure that the substrate was present at the same concentration in each assay. For each assay, we performed three independent experiments (two or three replicates each time) to be certain of the result.

2) Additionally, how do the authors know which position on the xylose ring is phosphorylated? Is this based on the previously published papers, if so, please clarify.

Author's response:

Koike et al. showed that Fam20B phosphorylates the xylose at the C2 hydroxyl position in the glycosaminoglycan-protein linkage region (*Biochem J* 421, 157-62 (2009); Figure 1). We have made proper reference to this point in the revised manuscript (Line 76), and we thank Dr. Eyers for carefully reading our manuscript!

Transparent review process: This review was submitted by Prof Patrick Eyers, University of Liverpool, and he was the sole reviewer.

Reviewer #2 (Remarks to the Author):

The authors present structural and biochemical data on the Fam20 protein kinases, an interesting and important family of kinases responsible for the phosphorylation of secreted proteins and proteoglycans. The detailed structural analysis builds on the authors' earlier work and rationalises many of their previous findings; while providing new insight into the function as well as the evolutionary history of the protein family.

The manuscript is carefully prepared, logical and well written, but could benefit from a few minor changes, as detailed below:

1. Page 3, line 49: Change "sequence similarity 20" to "sequence similarity of 20 %".

Author's response:

"Family with sequence similarity 20" was the name of this family, assigned by the Human Genome Organization Gene Nomenclature Committee (Nalbant et al. *BMC Genomics* 6, 11 (2005)). For clarity, we have put this term in the quotation marks in the revised manuscript (Line 49).

2. Page 4, line 61: Change FAM20A to fam20A.

3. Page 5, line 81: Change FAM20B to fam20B.

Author's response:

These were changed as suggested.

4. Page 6, line 105: K β 3-K α 3 nomenclature is not in agreement with traditional kinase secondary structure nomenclature. If this is common nomenclature for the FAM20 family? Please specify in the text or adjust the nomenclature accordingly.

Author's response:

The Fam20 proteins (especially Fam20A and Fam20C) contain long N-terminal segments in addition to the kinase domain. To adopt the nomenclature of traditional kinases, we use "K" to denote the secondary structures in the kinase domain (for example, the K β 3-K α 3 loop corresponds to the β 3- α 3 loop of canonical kinases), and "N" to denote the secondary structures in the N-terminal region. We used this nomenclature system in a paper we published previously (Cui, J. et al. *Elife* 6, e23990 (2017)). We have explained the nomenclature in the revised manuscript (Line 106).

5. Figure 2: Quantification of autoradiography data should be displayed, as a bar graph. This would be useful to more easily compare activity data between the mutants. Furthermore this data should be normalised to account for differing protein levels (Fig. 2C, 2D).

Author's response:

Quantification and normalization of Fig. 2C are performed and included as a bar graph. We did not quantify Fig. 2D because unlike Fig. 2C, the phosphorylation of OPN by overexpressed Fam20A/Fam20C in Fig. 2D is assessed by its change in electrophoretic mobility on SDS-PAGE, which is a bit difficult to quantify since some samples displayed "smearing". Nevertheless, we think that the result is clear based on the gel shift extent of OPN. We have used this assay in several papers to assess Fam20C activity (Tagliabracci et al. *Science* 336, 1150-3 (2012); Cui, J. et al. *Elife* 4, e06120 (2015)).

6. Page 9, line 75: Comparison between the Fam20A homodimer and Fam20C heterodimer interface would be clearer if the two interfaces were overlaid on a single, figure (an additional figure added to the SI).

Author's response:

A new figure is added as suggested (Supplementary Fig. 4B).

7. Page 10, line 207: E299 and I300 are discussed in the text but are not highlighted in Fig. 4.

Author's response:

N298 precedes E299 and I300 in Fam20A. The reason that we mutated E299-I300 to G-S is to generate an N-x-S consensus sequence for N-linked glycosylation of N298, in order to hinder Fam20A-Fam20C interaction in the tetramer. Since E299 and I300 do not play significant roles in the tetramer interface, we did not show them in Fig. 4 for the clarity of the figure.

8. Page 11, line 233: Is aqFam20 actually a larger protein containing more amino acids compared with the other family members or does it just behave as a larger protein on SDS-PAGE, SEC etc? This should be addressed in the text or at least in Fig. S2 (where aqFAM is not included).

Author's response:

aqFam20 is a slightly large protein (388 amino acids, see the Methods section) than hmFam20 (361 amino acids) and human Fam20B (348 amino acids). All three proteins contain two N-linked glycosylation sites, so they migrate more slowly compared to normal proteins on the gel. Nevertheless, aqFam20 it is still a monomer as judged by the elution

positions of the size exclusion molecular weight standards. We have included these information in the modified legend of Supplementary Fig. 3D. We did not include aqFam20 in Supplementary Fig. 2 because that figure is already very crowded, so we only included several representative sequences from the three Fam20 subfamilies.

9. There is much discussion on how the active site in Fam20B makes up the dimer interface of Fam20C, rendering it inaccessible to substrate, but not much discussion on where the active site of Fam20C is with respect to Fam20B. A figure of Fam20C and Fam20B overlaid with their respective catalytic sites highlighted, and a few sentences addressing this point would be of interest to the reader.

Author's response:

A new figure is added as suggested (Figure 6A), and we have expanded on this section in the main text (Page 12, Lines 248-258).

10. Page 14, line 302: How do the face-to-face dimers of FAM20 kinases compare to other the other pseudo-kinase/kinase heterodimers that are mentioned in the text? An additional supplementary figure should be included to highlight common and diverse features of these dimeric arrangement.

Author's response:

A new figure is included in the revised manuscript as suggested (Supplementary Fig. 6).

11. Fig. S2: The black box surrounding certain residues is undefined in the figure legend.

Author's response:

We have consolidated the legend of this figure in the revised manuscript.

REVIEWERS' COMMENTS:

Reviewer #1 (Remarks to the Author):

Through appropriate additional experiments and much clearer referencing, the authors have responded to all the comments that I suggested. The study is now suitable for publication, in my opinion. Through appropriate additional experiments and much clearer referencing, the authors have responded to all the comments that I suggested. The study is now suitable for publication, in my opinion, and especially since additional comments from Reviewer 2 have also been incorporated.